# Potential for Omega-3 Fatty Acids to Protect against the Adverse Effect of Phytosterols: Comparing Laboratory Outcomes in Adult Patients on Home Parenteral Nutrition Including Different Lipid Emulsions

**DOI:** 10.3390/biology11121699

**Published:** 2022-11-24

**Authors:** Sylwia Osowska, Marek Kunecki, Jacek Sobocki, Joanna Tokarczyk, Krystyna Majewska, Magdalena Burkacka, Marek Radkowski, Magdalena Makarewicz-Wujec, Helena L. Fisk, Sultan Mashnafi, Sabine Baumgartner, Jogchum Plat, Philip C. Calder

**Affiliations:** 1Applied Pharmacy Department, Warsaw Medical University, 02-097 Warsaw, Poland; 2Centre of Clinical Nutrition, Pirogow Hospital, 90-531 Lodz, Poland; 3Department of General Surgery and Clinical Nutrition, Centre for Postgraduate Medical Education, 00-416 Warsaw, Poland; 4Department of Immunopathology, Warsaw Medical University, 02-097 Warsaw, Poland; 5Institute of Pharmaceutical Care, University of Economics and Human Sciences, 01-043 Warsaw, Poland; 6School of Human Development and Health, Faculty of Medicine, University of Southampton, Southampton SO16 6YD, UK; 7Department of Nutrition and Movement Sciences, NUTRIM School of Translational Research in Metabolism, Maastricht University, 6200 MD Maastricht, The Netherlands; 8NIHR Southampton Biomedical Research Centre, University Hospital Southampton NHS Foundation Trust and University of Southampton, Southampton SO16 6YD, UK

**Keywords:** lipid emulsion, fish oil, phytosterol, parenteral nutrition, liver, inflammation

## Abstract

**Simple Summary:**

The choice of lipid emulsions (LEs) used in parenteral nutrition (PN) is based on fatty acid composition and phytosterol content. Phytosterols are believed to be detrimental in patients receiving PN. Data from this observational study suggest that the adverse effect of phytosterols delivered to home PN patients is mitigated by long-chain omega-3 fatty acids.

**Abstract:**

Background: the effect on liver function markers and inflammation of the different content of phytosterols in lipid emulsions (LEs) used in the parenteral nutrition (PN) regimen of adult home PN (HPN) patients is not clear. Methods: plasma phytosterol and cytokine concentrations, fatty acid composition, liver function markers, and triglycerides were measured in 58 adult HPN patients receiving one of three different LEs (soybean oil-based: Intralipid; olive oil-based: ClinOleic; containing fish oil: SMOFLipid). Results: patients receiving Intralipid had higher plasma campesterol and stigmasterol concentrations than those receiving ClinOleic or SMOFLipid. Plasma sterol concentrations were not different between patients receiving ClinOleic and SMOFLipid. Differences in plasma fatty acids reflected the fatty acid composition of the LEs. Markers of liver function did not differ among the three groups. Blood triglycerides were higher with ClinOleic than with Intralipid or SMOFLipid. Total bilirubin correlated positively with the plasma concentrations of two of the phytosterols, ALT correlated positively with one, AST with one, and GGT with three. Conclusions: liver function markers correlate with plasma plant sterol concentrations in adult HPN patients. Adult HPN patients receiving SMOFLipid are more likely to have liver function markers and triglycerides within the normal range than those receiving ClinOleic or Intralipid. The omega-3 fatty acids in SMOFLipid may act to mitigate the adverse effects of plant sterols on liver function.

## 1. Introduction

In parenteral nutrition (PN), lipid emulsions (LEs) are an important source of energy and the only source of essential fatty acids [1]. Depending on the oil from which they are produced, LEs differ in the amount and type of fatty acids [2]. The latter have a direct impact on metabolism, immune and inflammatory processes, and cell function [3]. Most of the LEs that are used in PN contain one or more vegetable oils. These oils contain plant sterols (phytosterols) [4,5]. Home parenteral nutrition (HPN) is an established therapy that aims to provide adequate amounts of all nutrients and water in order to prevent malnutrition in patients requiring long-term PN due to prolonged gastrointestinal tract failure [1,6,7]. One of the complications of long-term PN is liver damage [8]. Its etiology, which is believed to be multifactorial, is not yet fully understood [8]. However, the literature suggests that there may be two important LE-related factors: the presence of phytosterols which have a detrimental effect and the presence of different fatty acids, with a view that omega-6 fatty acids are detrimental and omega-3 fatty acids are protective [9,10,11,12]. Fish oil is a source of the bioactive omega-3 fatty acids eicosapentaenoic acid (EPA) and docosahexaenoic acid (DHA) [13]. In the pediatric population, many studies describe the prevention or even the reversal of liver damage by using fish oil-based LEs [14,15,16,17]. It is important to note, however, that infants are more likely to show cholestasis, while liver steatosis is more common in adults, although these phenomena are poorly understood [8,18].

Different LEs may be used as part of the nutrition support of adult HPN patients. As mentioned above, these LEs differ in content and composition of sterols, including plant sterols, and in composition of fatty acids. These differences between LEs might affect inflammation, lipid metabolism and liver function. Based upon findings in pediatric patients, we hypothesized that the inclusion of fish oil in a LE used as part of the support of adult patients on HPN will result in a better profile of liver function markers and less inflammation, and that these effects will be related to differences in plasma phytosterols. Therefore, the aim of this study was to compare plasma sterol concentrations in adult HPN patients receiving one of three different LEs (soybean oil-based: Intralipid; olive oil-based: ClinOleic; containing fish oil: SMOFLipid) and to investigate their relationship with markers of liver function and inflammation.

## 2. Materials and Methods

### 2.1. Study Design, Patients, and Interventions

This was a cross-sectional observational study with 3 groups of patients from 2 Polish parenteral nutrition centers (Department of Clinical Nutrition and Surgery, Orlowski Hospital in Warsaw and Center of Clinical Nutrition, Pirogov Hospital in Lodz). The study protocol was approved by the Bioethical Committee of Warsaw Medical University. In total, 58 stable patients with intestinal failure supported by HPN (33 women and 25 men; mean age 58 years) were recruited. Patient inclusion criteria were: age > 18 years; being part of the hospital’s HPN program; duration of HPN for a minimum of 2 years prior to the study on the same lipid emulsion; PN provided as 7 infusions per week; oral feeding and drug therapy unchanged during the 2 months prior to inclusion in the study; clinical stability. Exclusion criteria were: active infection in the last 12 months; liver or renal failure or both; pregnancy.

Each patient was prescribed indexed amounts of energy, macronutrients, fluids, and electrolytes in relation to their clinical condition, biochemical results, and standard recommendations. Patients could eat ad libitum and PN support had been adjusted individually over time in order for them to achieve their optimal weight, to neither gain nor lose weight, and to keep their biochemical results stable. Thus, PN support was tailored to meet patients’ needs. This approach is consistent with the ESPEN guidelines, which state “we recommend that the protein and energy requirements for chronic intestinal failure patients are based on individual patient characteristics and specific needs and the adequacy of the regimen is regularly evaluated through clinical, anthropometric and biochemical parameters” [19]. Oral intake provided about 500 kcal/day and a low-fat diet was recommended. Vitamins and trace elements were provided at one vial per day as recommended in stable HPN patients, and all patients received oral vitamin D supplementation (75 μg as cholecalciferol/day). Electrolytes and fluids were prescribed in relation to the biochemical results. All patients received comparable amounts of amino acids (0.7 to 1.0 g/kg per day or 50 to 52 g/day) and glucose (3.5 to 4.6 g/kg per day or 220 to 240 g/day) by the parenteral route. All patients received 20 g of lipid from the LE daily (i.e., 100 mL of emulsion); lipid provision was not adjusted for body weight. PN provided approximately 1300 kcal/day, with lipids providing about 15% of this. The ESPEN guidelines state that “many stable patients on HPN are satisfactorily maintained on 20–35 kcal total energy per kg per day” [19], which is consistent with our approach. Furthermore, our provision of lipid is consistent with the ESPEN guidelines to avoid essential fatty acid deficiency [6,19]. The non-protein calories to nitrogen ratio was maintained in the reference range of around 140. PN was administered by central catheter (Broviac) over 16–18 h per 24 h. Patients were receiving ClinOleic (80:20 olive oil:soybean oil; Baxter Healthcare, Maurepas, France), SMOFLipid (30:30:25:15 soybean oil:medium-chain triglycerides:olive oil:fish oil; Fresenius-Kabi, Bad-Homburg, Germany) or Intralipid (soybean oil; Fresenius-Kabi, Bad Homberg, Germany) as part of their routine nutrition support; these all contain 20 g of lipid per 100 mL. The LE provided to each patient was the clinician’s decision and was not guideline-driven. Due to differences in the vitamin E content of the different LEs, the daily parenteral dose of tocopherol was: 0.087 μmol in the Intralipid group, 0.075 μmol in the Clinoleic group, and 0.5 μmol in the SMOFlipid group.

The characteristics of the 3 groups are summarized in Table 1. All patients had comparable small bowel length (remaining intestine was 30 to 35%). The clinical heterogeneity of the patients studied reflects the clinical reality of patients for whom HPN is indicated. Blood samples were collected between 2019 and 2021.

### 2.2. Blood Processing and Overview of Analyses Performed

Blood was collected into disodium EDTA as anti-coagulant, 2–3 h after completing infusion of PN (lasting for 16 h). An aliquot was used for routine biochemical analyses. The following were measured: total bilirubin, alanine aminotransferase (ALT), aspartate aminotransferase (AST), gamma-glutamyltranspeptidase (GGT), total triglycerides, and C-reactive protein (CRP). An aliquot of blood was immediately centrifuged and plasma was isolated; this was stored at −80 °C until analysis. The following were measured in plasma: cholesterol, cholestanol, lathosterol, campesterol, stigmasterol, sitosterol, cytokines including interleukin (IL)-6, IL-8, IL-10, tumor necrosis factor (TNF)-α, interferon (IFN)-γ, and fatty acids. The concentrations of cholesterol, cholestanol, lathosterol, campesterol, stigmasterol, and sitosterol were also measured in original bottles of the LEs.

### 2.3. Measurement of Fatty Acids in Plasma

Lipid was extracted from plasma using 5 mL of chloroform:methanol (2:1; vol/vol) containing 0.2 M butylated hydroxytoluene as antioxidant. Sodium chloride (1 M; 1 mL) was added and the sample vortexed and then centrifuged. The lower solvent phase containing the lipid was aspirated and evaporated to dryness under nitrogen at 40 °C. Fatty acids were removed from complex lipids and simultaneously derivatized to methyl esters by incubation with 1 mL 2% H_2_SO_4_ (vol/vol) in methanol for a minimum of 2 h at 50 °C to form fatty acid methyl esters. The samples were then neutralized and fatty acid methyl esters transferred into hexane for analysis by gas chromatography. Fatty acid methyl esters were separated on a BPX-70 fused silica capillary column (30 m × 0.2 mm × 0.25 µm, manufactured by SGE) in a HP6890 gas chromatograph fitted with a flame ionization detector. Gas chromatography run conditions were as described elsewhere [20]. A Supelco^®^ 37 Component FAME Mix was used as a calibration reference standard (Sigma-Aldrich, Irvine, UK). FAME peaks were identified and integrated using Chem Station software (Agilent, Santa Clara, CA, USA) and fatty acid data are expressed as weight % of total fatty acids present.

### 2.4. Measurement of Plasma Cytokine Concentrations

The concentrations of TNF-α, IL-1β, IL-6, IL-8, IL-10, and IFN-γ were measured in plasma using a high sensitivity Bio-Techne multiplex immunoassay (R&D Systems, Abingdon, UK). Reagents were brought to room temperature before use and dilutions were prepared immediately before use according to the manufacturer’s instructions. Samples were read using a Bio-Rad-plex Luminex Analyzer. Data are expressed as pg/mL plasma.

### 2.5. Measurement of Sterol Concentrations

As internal standards, 5α-cholestane and epicoprostanol were added to plasma (or LE) samples, and these samples plus standards were saponified with 90% ethanolic sodium hydroxide for 1 hr at 60 °C. After 2 rounds of cyclohexane extraction, samples were derivatized with TMS reagent (pyridine, hexamethyldisilazane, and trimethylchlorosilane (9:3:1, vol/vol/vol)). Derivatized sterols were separated on a DB-XLB capillary column (30 m × 0.25 mm × 0.25 μm; Agilent Technologies, Amstelveen, Netherlands) in an HP6890 plus gas chromatograph fitted with a flame ionization detector. Gas chromatography run conditions were as described elsewhere [21]. Peaks were identified and integrated using Open Lab CDS Chem Station software (Agilent) and sterol concentrations were calculated relative to the internal standard 5α-cholestane concentration.

### 2.6. Statistical Analysis

Data were checked for normality using the Kolmogorov–Smirnov test. Much of the data were skewed and therefore all data are expressed as median and interquartile range. Comparisons were made across treatment groups using the Kruskal–Wallis test. Where the Kruskal–Wallis test was significant, pairwise comparisons between groups were conducted and *p* values were Bonferroni-adjusted for multiple comparisons. Correlations were investigated as Spearman rank correlations and are reported as Spearman’s ρ. Percentages were compared between groups using the chi-squared test. Statistical analyses were performed using SPSS version 21. In all cases, a value for *p* < 0.05 was taken to indicate a statistically significant difference.

## 3. Results

### 3.1. Sterol and Stanol Concentrations in the Lipid Emulsions and in Plasma

The sterol concentrations in the three LEs are shown in Table 2. The emulsions differed in total sterol (the sum of cholesterol, cholestanol, lathosterol, campesterol, stigmasterol, and sitosterol) content (ClinOleic 27.65 mg/dL, Intralipid 68.34 mg/dL; SMOFLipid 61.21 mg/dL); thus, patients in the ClinOleic group received less total sterols than those in the other two groups. Plant sterols (i.e., excluding cholesterol, cholestanol, and lathosterol) were higher in Intralipid (40.22 mg/dL) than in ClinOleic (22.14 mg/dL) and SMOFLipid (18.63 mg/dL); thus, patients in the ClinOleic and SMOFLipid groups received fairly similar amounts of phytosterols and these were less than the amount received by patients in the Intralipid group. Furthermore, the content of the different sterols differed across the emulsions. The most common sterol in ClinOleic was sitosterol, followed by cholesterol. In Intralipid, the most common sterols were cholesterol followed by sitosterol; there were also significant concentrations of stigmasterol and campesterol in Intralipid. In SMOFLipid, cholesterol was the most common sterol present, and there was also a high content of sitosterol.

Table 3 shows the sterol concentrations in the plasma of patients receiving the different lipid emulsions. Cholesterol concentrations were much higher than the concentrations of other sterols measured (Table 3). Cholestanol and lathosterol are markers of cholesterol absorption and endogenous cholesterol synthesis, respectively. Campesterol, stigmasterol, and sitosterol are plant sterols. Patients in the Intralipid group had higher plasma concentrations of campesterol and stigmasterol than those in the ClinOleic and SMOFLipid groups (Table 3); this is consistent with Intralipid containing higher amounts of these two phytosterols (Table 2). Furthermore, patients in the Intralipid group tended to have had higher plasma concentrations of sitosterol than those in the ClinOleic and SMOFLipid groups (Table 3). Plasma sterol concentrations were not different between the ClinOleic and SMOFLipid groups; this is consistent with the similar phytosterol content and composition of these two LEs.

### 3.2. Plasma Fatty Acids

The fatty acid compositions of Intralipid, ClinOleic, and SMOFLipid are described elsewhere [2] and will be summarized here. Because it is based solely on soybean oil, Intralipid is rich in linoleic acid (18:2n-6), which comprises about 53% of fatty acids present. Intralipid also contains about 8% α-linolenic acid (18:3n-3). ClinOleic is rich in oleic acid (18:1n-9) and contains about 19% linoleic acid and about 2% α-linoleic acid. SMOFLipid also contains about 19% linoleic acid and 2% α-linolenic acid, but it also contains EPA (about 3%) and DHA (about 2%).

Table 4 shows the plasma fatty acid composition according to the LE received. There were a number of significant differences between the groups. Plasma oleic acid was higher in the ClinOleic group than in the other two groups and was lower in the Intralipid group than the other two groups. Plasma linoleic and α-linolenic acids were higher in the Intralipid group than in the other two groups. Plasma arachidonic acid was lower in the SMOFLipid group than in the ClinOleic and Intralipid groups. Plasma EPA and DHA were both higher in the SMOFLipid group than in the other two groups. In general, these findings reflect the fatty acid composition of the emulsions themselves.

### 3.3. Plasma Liver Function Markers and Triglycerides

Table 5 shows the plasma liver function markers and triglycerides in the three groups. Liver function markers did not differ among groups. Triglycerides were significantly higher in the ClinOleic group than in the other two groups.

The % of patients with values for liver function markers and plasma triglycerides above the normal range is shown in Table 6. The % of patients with elevated ALT was highest in the ClinOleic and SMOFLipid groups, while the % with elevated AST was highest in the ClinOleic and Intralipid groups. The % of patients with elevated GGT was highest in the Intralipid group. The % of patients with elevated triglycerides was significantly higher in the ClinOleic group than in the other two groups.

### 3.4. Plasma Markers of Inflammation

Table 7 shows the plasma markers of inflammation in the three groups. CRP was lower in the ClinOleic group than in the other two groups, while IL-8 was higher in the ClinOleic than the Intralipid group.

### 3.5. Correlations between Liver Function Markers and Plasma Sterols and Stanols

Using data from all patients, irrespective of the type of LE they were receiving, bilirubin was positively correlated with plasma stigmasterol and sitosterol (ρ = 0.264, *p* = 0.032 and ρ = 0.290, *p* = 0.020, respectively) with a trend to a positive correlation with plasma campesterol (ρ = 0.236, *p* = 0.061). ALT and AST were both positively correlated with plasma sitosterol (ρ = 0.356, *p* = 0.004 and ρ = 0.412, *p* = 0.001, respectively). There was also a trend towards a positive correlation between AST and plasma stigmasterol (ρ = 0.233, *p* = 0.064). GGT was positively correlated with plasma cholestanol (ρ = 0.325, *p* = 0.009), campesterol (ρ = 0.42, *p* = 0.001), and sitosterol (ρ = 0.502, *p* < 0.001). Figure 1 shows these associations.

When correlations between liver function markers and plasma sterols were investigated within each LE group, there were no significant correlations in either the Intralipid or SMOFLipid groups. However, in the ClinOleic group, ALT and GGT were both positively correlated with plasma stigmasterol, sitosterol, and campesterol, while bilirubin and AST were both positively correlated with sitosterol and stigmasterol.

### 3.6. Relationship between Plasma EPA and GGT

Figure 2 shows an inverse relationship between plasma EPA and GGT, although this was not statistically significant.

## 4. Discussion

The main findings of our study suggest that provision of bioactive omega-3 polyunsaturated fatty acids (EPA and DHA) might attenuate the deleterious effects on liver health of phytosterols present in plant-based LEs used in patients on long-term PN. Surprisingly, liver function markers in patients in the Intralipid group, who received the highest amount of phytosterols and whose plasma concentrations of campesterol and stigmasterol were significantly higher than in those receiving Clinoleic or SMOFlipid, were not different from patients in the other two groups. At the same time, the plasma level of one omega-3 fatty acid, namely, α-linolenic acid (18:3n-3), was significantly higher in the Intralipid group than in the other two groups, and the plasma level of another omega-3 fatty acid, EPA, was significantly higher than in the Clinoleic group. That might suggest a protective effect of omega-3 fatty acids on liver function in adult HPN patients. This is further emphasized by comparison of findings between patients receiving ClinOleic and SMOFLipid; the phytosterol content of these two LEs is lower than in Intralipid, but they differ in the content of omega-3 fatty acids. Although patients in the ClinOleic and SMOFLipid groups received similar amounts of the different phytosterols and had plasma sterol concentrations that did not differ, those in the ClinOleic group tended to be more likely to have concentrations of bilirubin, AST, GGT, and triglycerides above the normal range than patients in the SMOFLipid group. Furthermore, in the ClinOleic group, there were significant correlations between plasma phytosterol concentrations and all of the liver function markers; these correlations were not seen in patients in the SMOFLipid group. This suggests that the adverse relation between phytosterols and liver function might be attenuated by SMOFLipid. In support of this, there was an inverse association of plasma EPA with GGT, although this did not reach statistical significance, perhaps because of the sample size. It is important to note that there were also no significant correlations between plasma phytosterols and liver function markers in the Intralipid group, despite Intralipid containing more phytosterols than the other LEs and despite patients receiving Intralipid having the highest plasma phytosterol concentrations. This unexpected observation may relate to the low lipid load used in the current study: patients received 20 g lipid daily from PN, which equates to <0.3 g/kg body weight for a 70 kg individual. Thus, in the current study, Intralipid may have been used at a dose that is below the dose at which it adversely affects liver function.

Several studies have shown the reversal of cholestasis in infants receiving PN, either by decreasing the dose of soybean oil-based LEs [22,23] or by administration of pure fish oil-based LEs or a mixture of different lipids that included fish oil [24]. Recommendations for lipids in PN in preterm and term infants are 3–4 g/kg per day, and in children are a maximum of 3 g/kg day [25]. These greatly exceed recommendations for adults receiving HPN (0.7 to 1.3 g/kg per day) [6] and the lipid dose used in the current study, making direct comparisons between findings in infants/children and adults difficult. Furthermore, infants are more likely to show cholestasis than adults, while in adults, steatosis is more common [8,18].

Many factors can lead to liver injury in patients receiving long-term PN. These include high doses of glucose, insufficient trace elements, and the presence of sepsis. These factors are not likely to be relevant to the differences between the patients studied here because they all received similar amounts of glucose and trace elements and there was no recent sepsis. The mechanisms of liver injury during long-term PN that are currently receiving the most attention include the deleterious effect of plant sterols present in plant-based LEs and the pro-inflammatory effect of omega-6 polyunsaturated fatty acids. The concentrations of cholesterol, campesterol, stigmasterol, and sitosterol we report for Intralipid and ClinOleic are consistent with the concentrations reported by Forcielli et al. [4], while the total concentrations of phytosterols we report for Intralipid (22.2 vs. 20.8 mg/dL) and ClinOleic (40.2 vs. 42.2 mg/dL) are consistent with the report of Llop Talaverón et al. [5], but our value for SMOFLipid is higher than theirs (18.6 vs. 12.4 mg/dL). This might reflect batch differences, as reported by others [5]. The plasma concentrations of phytosterols reflected the phytosterol content of the LEs, as might be expected: plasma campesterol and stigmasterol were higher in patients receiving Intralipid. In a study with mouse hepatocytes, out of three phytosterols tested (stigmasterol, campesterol, and sitosterol), stigmasterol proved to have the greatest potential in promoting cholestasis through antagonism of multipurpose fanesoid X receptor (FXR) function and reduction in expression of the canalicular bile acid transporter (ABCB11) [26]. Increased serum stigmasterol was correlated with liver inflammation and cholestasis in children receiving PN [27]. In the present study, sitosterol was positively correlated with plasma levels of bilirubin, ALT, AST, and GGT. Bilirubin was also positively correlated with stigmasterol, with a trend to positive correlation with plasma campesterol. GGT was positively correlated to cholestanol and campesterol. These correlations were seen only in patients receiving ClinOleic. This suggests that the fatty acid composition of LEs influences the effects of phytosterols on liver function. This might relate to the differential effects of fatty acids on inflammation. It is important to note that the LEs used here also differ in their content of tocopherol, and that might also impact inflammation.

Proinflammatory cytokines lead to suppression of nuclear receptor-mediated gene expression in the liver, including FXR-dependent pathways, which, as a consequence, leads to cholestasis [28,29,30]. In the current study, a significantly higher plasma concentration of IL-8 was observed in the ClinOleic group in comparison to the other two groups. The mechanism behind this is not clear. However, IL-8 production has been shown to be enhanced by omega-6 fatty acids and by arachidonic acid metabolites [31,32]. Intralipid contains more omega-6 fatty acids (as linoleic acid) than ClinOleic and, therefore, might be expected to result in higher IL-8 concentrations, but this was not seen. ClinOleic contains the highest concentration of oleic acid, which was reflected in the plasma of the patients. This emulsion contains 20% soybean oil in comparison to 30% soybean oil present in SMOFlipid. This difference in soybean oil content did not result in a different plasma concentration of linoleic acid. Plasma arachidonic acid was not different between the ClinOleic and Intralipid groups, but was higher than in the SMOFLipid group. Furthermore, the ClinOleic group had lower plasma EPA than both the Intralipid and SMOFLipid groups. The ratio of EPA to arachidonic acid was lowest in the ClinOleic group (0.094) compared with the Intralipid (0.135) and SMOFLipid (0.383) groups. EPA has anti-inflammatory and inflammation-resolving actions [33], arachidonic acid is linked to potential for increased inflammation [34], and these two fatty acids act to oppose one another’s action [35]. Therefore, the ratio between these two fatty acids may be the link between the different LEs and inflammation.

Patients in the ClinOleic group had a significantly higher plasma level of triglycerides than in the other two groups, and these were more likely to be above the reference value. This could be due to ClinOleic having the lowest content of polyunsaturated fatty acids. Polyunsaturated fatty acids are strong activators of peroxisome proliferator activated receptors (PPARs), especially PPAR-α, with DHA being the strongest fatty acid activator [36]. PPAR-α plays a key role in the regulation of hepatic fatty acid oxidation by increasing the expression of the fatty acid transport protein, fatty acid translocase, acyl-CoA oxidase, and carnitine palmitoyltransferase [37]. These effects act to partition fatty acids towards oxidation and away from triglyceride synthesis [38,39]. Furthermore, PPAR-α amplifies the expression of lipoprotein lipase and inhibits apolipoprotein C-III synthesis [40]. These mechanisms together result in decreased hepatic accumulation and secretion of triglycerides, and decreased blood triglyceride concentrations, and might explain why plasma triglycerides are lower in patients receiving more polyunsaturated fatty acids (i.e., Intralipid and SMOFLipid).

There may be another factor involved in determining the different plasma triglyceride concentrations in patients receiving the different LEs. Of the three LEs studied here, ClinOleic is the only one in a plastic container. An interaction between the container and lipid stability [41], and a link between plastic containers and a higher incidence of hypertriglyceridemia have been described [42].

It is important to note that this study has some limitations. Firstly, patients were not randomly allocated to receive the different LEs, and this could introduce a bias. Secondly, the number of patients studied is modest, and this is most likely why apparent differences between groups in the percentage of patients with elevated liver function markers are not statistically significant. Thirdly, we did not consider the effect of differences in provision of tocopherol between the groups, which can influence inflammation. Fourthly, we have no data on liver histology. Finally, as this was a cross sectional study, causality cannot be inferred. Thus, the findings need to be interpreted with caution.

## 5. Conclusions

We conclude that phytosterol content and composition and fatty acid composition are important in determining the physiological impact of LEs used in adult HPN. Phytosterols are linked to impaired liver function, but we show here that this relationship might be attenuated by bioactive omega-3 fatty acids (EPA and DHA), most likely through their effects on inflammation and hepatic fatty acid and triglyceride metabolism.

## Figures and Tables

**Figure 1 biology-11-01699-f001:**
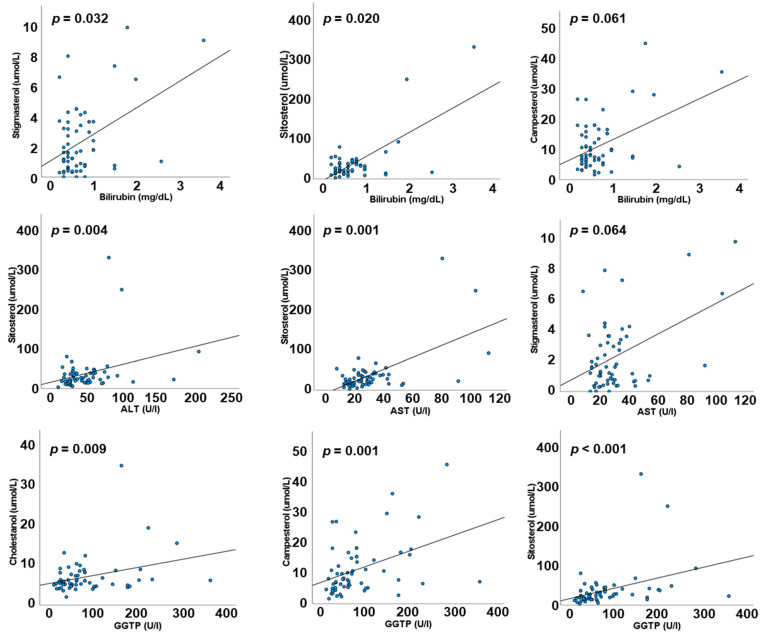
Correlations between liver function markers and plasma concentrations of phytosterols. Data are for all patients irrespective of lipid emulsion.

**Figure 2 biology-11-01699-f002:**
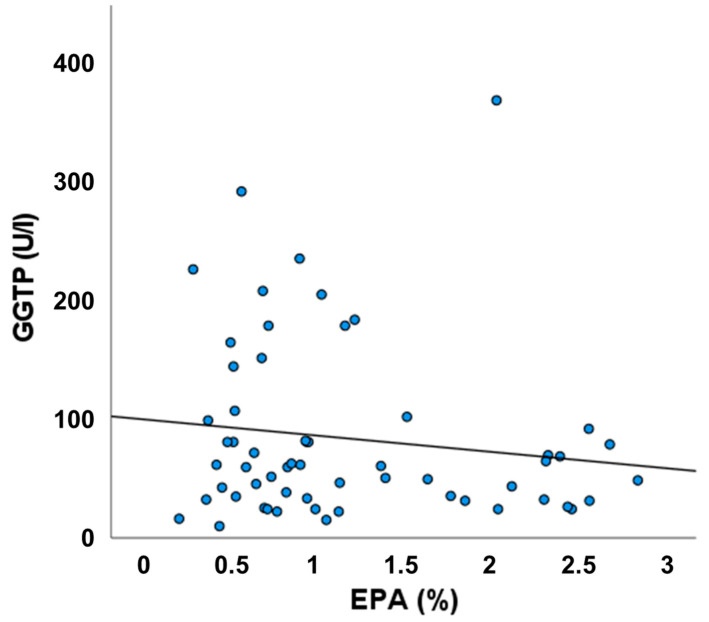
Association between plasma eicosapentaenoic acid (EPA) and GGT. Data are for all patients irrespective of lipid emulsion.

**Table 1 biology-11-01699-t001:** Characteristics of the patients according to lipid emulsion received.

	ClinOleic	SMOFLipid	Intralipid
Number of patients	21	17	20
Age range, years (mean)	19–91 (60.3)	27–84 (54.5)	25–89 (59.0)
TPN duration, months (mean)	26–72 (46.5)	24–40 (33.4)	32–78 (48.2)
Male (*n*)	8	7	10
Female (*n*)	13	10	10
Etiology of intestinal failure (*n*):			
Bowel obstruction	2	2	2
Mesenteric ischemia	5	3	3
Surgical complications	3	4	5
Crohn’s disease	3	3	4
Adhesion ileus	3	1	2
Radiation enteropathy	4	2	2
Malabsorption	1	2	2

**Table 2 biology-11-01699-t002:** Sterol and stanol concentrations (mg/dL) in the three lipid emulsions. Data are mean ± SD from three replicates.

Sterol or Stanol	ClinOleic	Intralipid	SMOFLipid
Cholesterol	5.37 ± 0.67	27.65 ± 1.14 *	42.00 ± 1.88 ^‡,¶^
Cholestanol	0.06 ± 0.02	0.22 ± 0.01 *	0.35 ± 0.02 ^‡,¶^
Lathosterol	0.08 ± 0.02	0.25 ± 0.01 *	0.23 ± 0.01 ^¶^
Campesterol	1.88 ± 0.22	7.05 ± 0.33 *	2.89 ± 0.10 ^‡,¶^
Sitosterol	18.31 ± 2.16	24.08 ± 0.76 *	12.46 ± 0.25 ^‡,¶^
Campestanol	0.06 ± 0.01	0.16 ± 0.02 *	0.07 ± 0.01 ^‡^
Stigmasterol	1.12 ± 0.15	7.44 ± 0.30 *	2.67 ± 0.07 ^‡,¶^
Sitostanol	0.77 ± 0.09	1.49 ± 0.03 *	0.54 ± 0.04 ^‡,¶^

Significant *p* values after adjustment for multiple comparisons: * < 0.01 Intralipid vs. ClinOleic; ^‡^ < 0.01 SMOFLipid vs. Intralipid; ^¶^ < 0.001 SMOFLipid vs. ClinOleic.

**Table 3 biology-11-01699-t003:** Plasma sterol concentrations in patients according to the lipid emulsion being received. Data are median (interquartile range).

Sterol or Stanol	ClinOleic	Intralipid	SMOFLipid
Cholesterol (mmol/L)	3.40 (2.65, 3.95)	2.94 (2.59, 3.33)	2.89 (2.36, 3.88)
Cholestanol (μmol/L)	5.45 (4.63, 6.51)	6.14 (4.87, 8.80)	6.44 (5.5, 8.22)
Lathosterol (μmol/L)	10.85 (7.51, 16.28)	11.64 (3.69, 14.81)	12.39 (6.59, 19.94)
Campesterol (μmol/L)	4.95 (3.19, 6.80)	15.17 * (9.99, 17.94)	7.13 ^‡‡^ (6.33, 9.68)
Sitosterol (μmol/L)	23.18 (13.5, 48.6)	34.2 (19.0, 42.2)	21.8 (15.0, 27.6)
Stigmasterol (μmol/L)	0.52 (0.31, 0.87)	3.55 * (2.13, 4.40)	1.58 ^‡^ (1.09, 1.76)

Significant *p* values after adjustment for multiple comparisons: * < 0.001 Intralipid vs. ClinOleic; ^‡^ = 0.048 SMOFLipid vs. Intralipid; ^‡‡^ = 0.023 SMOFLipid vs. Intralipid.

**Table 4 biology-11-01699-t004:** Plasma fatty acid composition (% of total fatty acids) in patients receiving different lipid emulsions. Data are median (interquartile range).

Fatty Acid	ClinOleic	Intralipid	SMOFLipid
Myristic (14:0)	1.04 (0.84, 1.32)	1.12 (0.92, 1.42)	1.13 (1.02, 1.40)
Palmitic (16:0)	25.97 (24.46, 27.35)	24.66 (23.85, 25.63)	25.31 (24.38, 28.73)
Palmitoleic (16:1n-7)	4.27 (2.27, 5.11)	3.79 (2.90, 4.23)	3.74 (3.03, 4.61)
Stearic (18:0)	7.34 (6.84, 8.08)	7.96 (6.88, 9.24)	7.67 (6.91, 8.80)
Oleic (18:1n-9)	31.34 (27.64, 33.38)	21.78 * (20.8, 23.51)	25.27 ^‡‡^ (23.84, 29.7)
Vaccenic (18:1n-7)	2.55 (2.13, 2.80)	2.15 * (1.96, 2.28)	2.46 (1.90, 2.67)
Linoleic (18:2n-6)	14.25 (12.01, 19.00)	22.67 ** (21.21, 26.08)	16.06 ^‡‡‡^ (12.99, 19.87)
α-Linolenic (18:3n-3)	0.42 (0.35, 0.50)	0.91 ** (0.73, 1.10)	0.60 ^‡‡^ (0.48, 0.71)
Dihomo-γ-linolenic (20:3n-6)	1.69 (1.37, 2.03)	1.86 (1.51, 2.16)	1.51 (1.17, 1.86)
Arachidonic (20:4n-6)	6.86 (6.07, 8.33)	7.03 * (5.85, 7.68)	5.77 ^‡,¶^ (5.13, 6.18)
Eicosapentaenoic (20:5n-3)	0.65 (0.45, 0.75)	0.95 * (0.69, 1.22)	2.21 ^‡‡,¶¶^ (1.62, 2.41)
Docosapentaenoic (22:5n-3)	0.54 (0.45, 0.64)	0.55 (0.45, 0.64)	0.88 ^‡‡‡,¶¶^ (0.69, 1.21)
Docosahexaenoic (22:6n-3)	1.61 (1.20, 2.11)	1.78 (1.41, 2.42)	3.52 ^‡‡‡,¶¶^ (3.04, 4.18)

Significant *p* values after adjustment for multiple comparisons: Intralipid vs. ClinOleic * < 0.05, ** < 0.001; SMOFLipid vs. Intralipid ^‡^ < 0.05, ^‡‡^
*p* < 0.01, ^‡‡‡^ < 0.001; SMOFLipid vs. ClinOleic ^¶^ = 0.033, ^¶¶^ < 0.001.

**Table 5 biology-11-01699-t005:** Plasma liver function markers and triglycerides in patients receiving different lipid emulsions. Data are median (interquartile range).

Marker	ClinOleic	Intralipid	SMOFLipid
Total bilirubin (mg/dL)	0.6 (0.4, 0.8)	0.6 (0.4, 0.9)	0.4 (0.3, 0.7)
ALT (U/L)	46 (27, 61)	36 (29, 59)	34 (26, 60)
AST (U/L)	28 (21, 43)	26 (21, 36)	25 (18, 32)
GGT (U/L)	50 (25, 101)	80 (35, 150)	61 (38, 75)
Triglycerides (mg/dL)	178 (114, 236)	94 * (83, 146)	111 ^‡^ (70, 148)

Significant *p* values after adjustment for multiple comparisons: * = 0.015 Intralipid vs. ClinOleic; ^‡^ = 0.035 SMOFLipid vs. Intralipid. Reference values: total bilirubin: 0.2–1.3 mg/dL; ALT: 14–59 U/L; AST: 14–36 U/L; GGT:12–43 U/L; triglycerides < 150 mg/dL.

**Table 6 biology-11-01699-t006:** Percentage of patients in each group with plasma liver function markers and triglycerides above the normal range.

Marker	ClinOleic	Intralipid	SMOFLipid
Total bilirubin (mg/dL)	9.5	10.0	5.8
ALT (U/L)	28.6	15.0	29.4
AST (U/L)	23.8	25.0	11.8
GGT (U/L)	29.0	55.0	23.5
Triglycerides (mg/dL)	52.4	15.0 *	11.8 ^¶^

Significant *p* values: * = 0.012 Intralipid vs. ClinOleic; ^¶^
*p* = 0.037 SMOFLipid vs. ClinOleic.

**Table 7 biology-11-01699-t007:** Plasma inflammatory markers in patients receiving different lipid emulsions. Data are median (interquartile range).

Marker	ClinOleic	Intralipid	SMOFLipid
CRP (mg/L)	4.10 (0.60, 5.95)	6.36 * (5.57, 10.00)	5.49 ^¶^ (5.01, 10.09)
IL-1β (pg/mL)	1.00 (0.54, 1.39)	0.96 (0.63, 1.39)	0.80 (0.43, 1.51)
IL-6 (pg/mL)	5.07 (1.94, 5.80)	2.99 (2.36, 4.93)	3.09 (2.16, 5.12)
IL-8 (pg/mL)	36.4 (10.2, 34.8)	9.6 ** (4.6, 12.3)	10.6 (5.3, 26.8)
IL-10 (pg/mL)	1.92 (0.88, 1.86)	1.90 (1.02, 2.70)	1.85 (1.32, 2.36)
IFN-γ (pg/mL)	2.59 (0.13, 3.88)	1.26 (0.66, 6.00)	1.12 (0.32, 2.23)
TNF-α (pg/mL)	19.6 (16.3, 21.9)	14.6 (12.5, 20.3)	16.0 (13.7, 18.9)

Significant *p* values after adjustment for multiple comparisons: * = 0.012 Intralipid vs. ClinOleic; ** = 0.002 Intralipid vs. ClinOleic; ^¶^ = 0.039 SMOFLipid vs. ClinOleic. Reference values for CRP are 0–10 mg/L.

## Data Availability

Data are available from the corresponding author.

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
