# Peer review of "Potential for Omega-3 Fatty Acids to Protect against the Adverse Effect of Phytosterols: Comparing Laboratory Outcomes in Adult Patients on Home Parenteral Nutrition Including Different Lipid Emulsions"

_biology, 2022, doi:10.3390/biology11121699_

Round 1

Reviewer 1 Report

In this paper, Osowska et al. investigated the potential of omega-3 fatty acids to protect against the adverse effect of phytosterols in cross-sectional observational study.

The sample size of 58 included patients is sufficient because short bowel syndrme is a rare disease. The topic is quite interesting and contemporary, since the damaging effect of lipids is still a matter of debate. The study is based on comprehensive data based on LFTs, inflammatory makers, plasma sterol and fatty acid composition. Moreover, the paper comprises detailed analysis and the topic is of broad interest for the readership of Biology.

The paper is succinctly written, although, while reading the manuscript I have some annotations:

1.    Figure 1: please add the p-values

2.    I did not get the meaning of the tagging superscripted letters – what is the difference between “a, b, c “? – would you mind explaining this better or change that to an easier labelling?

3.    The multifactorial effect of liver damage was mentioned in the text – a short paragraph with further contributors should be added (glucose, sepsis, trace elements etc.).

4.    The amount of amino acids and glucose in the patient population should be added in g/kg/body weight since the absolute content is not important

5.    Methods: is it correct that all patients received app. 1300 kcal PN per day regardless of their daily nedds

6.    Methods: The patients received 20 g in total lipids or per 100 ml – please clarify (line 103 page 3/14)

7.    Table 2: there is no benefit to include 2 different units – please delete one to focus on the results and please highlight the significant differences 

Reviewer 2 Report

This work offers an interesting piece to be fitted into the puzzle of IFALD, focusing on adult HPN patients. I find the article reasonably well written and the conclusion supported mostly by the results. There are some notions I would like to discuss, especially regarding the clinical aspects of the study cohort.

- The simple summary - first paragraph: I would say most experienced centers regard LE phytosterol content as well as FA.  Suggest to rephrase to introduce the importance of both FA and sterols, as in fact written in Introduction.

- Introduction: I suggest to add something on known differences between pediatric and adult IFALD: adults have more steatosis altogether so this difference is a known but not sufficiently understood phenomenon (e.g. Sasdelli AS, Agostini F, Pazzeschi C, Guidetti M, Lal S, Pironi L. Assessment of Intestinal Failure Associated Liver Disease according to different diagnostic criteria. Clin Nutr. 2019 Jun;38(3):1198-1205. doi: 10.1016/j.clnu.2018.04.019. Epub 2018 May 8. PMID: 29778510.) 

Methods: Suggest to more clearly point out that these three subgroups were clinically determined, and the clinician had made the choice of LE. Also suggest to add the general clinical guidelines you have followed in choosing the LE (or lack of thereof). Is there a chance that Intralipid group was in fact an "older" cohort indicating a change in other clinical management may also have changed during time? Therefore I suggest to add study recruitment time periods as well. One major omission is the lack of reported infections which have a profound effect on liver health in HPN patients. Can you provide such data? If not this should be explained.

- 15% Lipids from total energy content is quite a small amount, which you have mentioned in Discussion. How it is possible that every patient received exactly 20 g, -  there was no variation in this (in opposition to glucose and amino acids)?  

- was there no peroral vitamin supplementation?

Results:  You mention later in discussion the possibility of batch differences explaining differing results from previous works. Suggest to add purchase time or similar to help future research in this respect.

Tables are missing info on actual p-values, which I think should be added. The footnote explanation on p-values on rows on "sharing superscript letters" is not reader friendly.

Section 3.2: Suggest to rephrase the first sentence to clearly indicate if these percentages are results you have obtained.

DIscussion: It is not possible to use this kind of retrospective uncontrolled data to describe causality. We also know too little about other liver damaging factors, such as infections, other components of PN. Suggest to tone down the conclusion to take this into account. Associations are easier to support by this data. P 11 r 355: infant PN macronutrient dosing is based on energy needs which are always higher per kg - so this comparison is naturally not at all relevant here.  If any comparisons are needed, they should be based on %total PN energy. The reason why infants and young children tend to have IFALD presenting as cholestasis more often than adults remains unknown.

Lack of histopathological data on the liver disease is a major problem, since liver function tests are only a substitute of the actual liver affection.

Round 2

Reviewer 1 Report

In this paper, Osowska et al. presented the revised version of their manuscript. Unfortunately, they attached no cover letter (or maybe not in English?) – so there are still unanswered questions:

1.    Page 8/16: line 277 -  0.033 in this line is wrong – the p-value presentation is now more clear

2. Methods: is it correct that all patients received app. 1300 kcal PN per day regardless of their daily needs?

3. Methods: The patients received 20 g in total lipids per patients -  was the amount of lipids body weight adjusted?

Reviewer 2 Report

Thank you for the comments. 

I would still like to see the timeline of the study recruitment period in text. IF care has evolved greatly over recent years in other than just PN.

Accept the current article.
